# CROSS-LINGUAL LONG-TAILED ENTITY ALIGNMENT IN KNOWLEDGE GRAPHS

## ABSTRACT

Entity alignment (EA) models rely mostly on triples and structural information of Knowledge Graphs (KGs), but underperform on sparsely connected long-tailed entities. We address this gap by proposing a model, **ContrastEA**, that leverages pre-trained Language Models (LM), e.g. me5, to generate entity representations, followed by a novel contrastive learning approach that incorporates hard-negative mining strategies with *top-k* negatives per entity, alongside NT-Xent loss to separate challenging entity pairs. In addition, to address the under-representation of long-tailed entities in benchmark datasets, we curate a new dataset from DBpedia comprising long-tailed entities per language – Arabic, German, Portuguese, Italian, Hindi, Russian, and Japanese, each aligned to English (a total of 154,296 cross-lingual entity pairs). Our results demonstrate that ContrastEA outperforms the classic EA models on three benchmark datasets, improving Hits@1 by 6–20 percentage points, and achieves SOTA on the curated dataset over the long-tailed EA models.

## 1 INTRODUCTION

A Knowledge Graph (KG) is a structural network of relational facts stored in the form of triples ⟨*head entity, relation, tail entity*⟩ and serves as the backbone of various applications, including semantic search (Bast et al., 2016), question answering (Yasunaga et al., 2021), and recommender systems (Xian et al., 2019). Most KGs are created from single sources designed for specific purposes, representing niche domain knowledge, and therefore also with limited coverage. This limitation is further amplified in multilingual KGs (Kaffee et al., 2023), where the same entity may have more facts in one language than in another. The same entity may exist in different KGs in different forms. For example, dbr:Albert_Einstein in DBpedia (Lehmann et al., 2015) and wd:Q937 in Wikidata (Vrandečić & Krötzsch, 2014) refer to the same entity *Albert Einstein*. Therefore, integrating multiple KGs is the key to completeness and knowledge fusion for better inference and reasoning, as different KGs contain complementary knowledge.

Entity alignment (EA) is the core task of KG integration by finding entities across different KGs that refer to the same (real-world) entity. However, a principal challenge is the heterogeneity of the underlying schemas across different KGs. KG embedding based methods have proved to be an effective approach for EA (Chen et al., 2018; 2017; Pei et al., 2019a;b; Guo et al., 2019; Sun et al., 2020) and can be summarized into two categories: (i) the KG embedding model encodes the source and target KGs into two separate embedding spaces, and trains an alignment method with aligned initial seeds to learn the mapping between the same entities across KGs (Chen et al., 2018; 2017; Pei et al., 2019a); (ii) an alignment model is trained to embed the two KGs into one unified space using initial seed alignment (Li et al., 2018; Sun et al., 2017; 2018; Trisedya et al., 2019; Wang et al., 2018; Zhu et al., 2017). These models leverage the relational and structural information of the entities, assuming the KGs are strongly connected and equivalent entities have similar neighbourhood structures in different KGs. However, KGs are inherently sparse, with approximately 50% of the entities being *long-tailed*, having three connections or less to other entities (Guo et al., 2019). For instance, 54.7% of the entities in English DBpedia and 61.5% in the German DBpedia appear fewer than three times.

Although embedding-based EA models can potentially mitigate the challenge of heterogeneity, they inherently struggle with insufficient structural and neighbourhood information. This raises the open

question *How can we align long-tailed entities across KGs with fewer or no connections to their neighbouring entities?* In this work, we propose a novel long-tailed EA model **ContrastEA**, that harnesses pretrained Language Models (LMs) to generate entity name representations, followed by a contrastive learning framework to align the same entities across different KGs. A hard-negative mining strategy is proposed to select the most informative *top-k* negatives per entity, ensuring the model to learn to differentiate between entities that are deceptively similar to the actual target entity. **ContrastEA** also includes the NT-Xent loss, which ensures the closeness of the positive pairs (i.e., same entities in different KGs) while pushing away the negative pairs in the embedding space.

Recent work (Guo et al., 2019) also uncovers that existing EA benchmark datasets fail to reflect the sparsity of the real-world KGs. To this end, the SRPRS benchmark (Guo et al., 2019) was introduced, with a more pronounced presence of long-tailed entities. However, entities with zero connections to other entities are not included in the SRPRS dataset. To address this and truly reflect the sparseness of KGs, we curate a multilingual EA benchmark dataset, **LT-EA-25K**, from DBpedia containing both long-tailed entities and entities with no links at all.

Prior works on long-tailed EA have used the latent representation of entity names derived from pretrained word embeddings as an additional signal to the structural information (Zeng et al., 2020), while classical EA models used entity name embeddings to initialize the feature matrices for structural representations (Wu et al., 2019a;b; Xu et al., 2019). In contrast, our proposed model **ContrastEA** uses only entity names, without depending on structural or relational information. We show that **ContrastEA** achieves state-of-the-art results on both the existing benchmarks as well as our curated dataset, ensuring its robustness for entities with varying connectivity. Our main contributions are:

- We propose a novel contrastive learning based EA approach **ContrastEA** that incorporates hard-negative mining strategies with *top-k* negatives per entity. It also exploits the NT-Xent loss to separate challenging entity pairs.

- We propose a multilingual long-tailed EA benchmark dataset **LT-EA-25K**, which comprises 25,000 long-tailed entities per language – Arabic, German, Portuguese, Italian, Russian, and Japanese, each aligned to English (a total of 154,296 cross-lingual entity pairs) and 4,296 aligned pairs of Hindi–English [1].

- Our detailed evaluation demonstrates that ContrastEA is robust for popular, long-tailed and entities with zero connectivity. Our results show that ContrastEA outperforms the classic EA models on three benchmark datasets, improving Hits@1 by 6–20 percentage points, as well as on the curated dataset by 10 percentage points on Hits@1 over the long-tailed EA models.

## 2 METHODOLOGY

**Preliminaries**  Given a KG $G = (E, R)$, where $E$ and $R$ are the set of entities and relations, respectively, the notation $< s, p, o > \in T$ represents a triple belonging to the set of triples $T$ in the KG, where $(s, o) \in E$ are the head and tail entities, and $p \in R$ represents the relation between them. Additionally, the entities in a KG are related to facts that are literals in the form of text, images, numbers, etc. Given a source KG $G_1 = (E_1, R_1)$, a target KG $G_2 = (E_2, R_2)$ and $T_1$ and $T_2$ are the sets of triples in the source and target KG, respectively. For training, the seed entity pairs are given by $S = \{(s, t) \mid s \in E_1, t \in E_2, s \leftrightarrow t\}$, where $\leftrightarrow$ denotes equivalence, designating $s$ and $t$ as being the same entity (Zhao et al., 2020).

**Problem Statement**  We consider the task of long-tailed EA as a supervised contrastive learning problem. Given a training set $\mathcal{D} = \{(s_i, t_i)\}_{i=1}^N$ of $N$ aligned seed entities, where $s_i$ and $t_i$ are the $i^{\text{th}}$ source and target entity, respectively. The goal is to learn a parametrised encoder $f_\theta : \mathcal{X} \to \mathbb{R}^d$ to generate a $d-$dimensional vector for the entity names of both $s_i$ and $t_i$ and subsequently train a contrastive learning model for EA. The training objective of the alignment model serves to encourage $f_\theta(s_i)$ to be similar to $f_\theta(t_i)$ and dissimilar to $f_\theta(t_i)$, for $i \neq j$.

---

[1]The data is made available in the supplementary folder

## 2.1 ENTITY REPRESENTATIONS

As mentioned earlier, EA models that use structural, relational, or neighbourhood information of the entities are ill-suited for aligning long-tailed entities across KGs, as they frequently lack such information. Therefore, we focus on the most consistent and omnipresent feature: *entity names*. Entity descriptions and attributes are often unavailable for less connected entities, but names are informative and often sufficient to identify an entity (Yaghoobzadeh et al., 2018; Yaghoobzadeh & Schütze, 2017). We use a transformer encoder $f_\theta$ to map entity names to $d$-dimensional vectors. For an input text $x$, the encoder returns the hidden states $H \in \mathbb{R}^{m \times L \times d}$, where $L$ is the length after padding/truncation, and $m$ is the batch size. To obtain a single representation of the entity name, masked mean pooling is used where $M \in \{0, 1\}^{m \times L}$ and the pooled embedding of the $i^{\text{th}}$ entity is:

$$e_i = \frac{\sum_j M_{i,j} H_{i,j}}{\sum_j M_{i,j}} \in \mathbb{R}^d$$

Leveraging entity embeddings for EA instead of simple string matching enables effective cross-lingual entity alignment. These representations are crucial when the entity names differ across languages with different vocabularies, overcoming challenges of translation and transliteration.

## 2.2 CONTRASTIVE LEARNING

Our proposed **ContrastEA** model focuses on pulling together similar entities while pushing apart hard-to-distinguish entities, i.e., misleading similar entities, aiming for a more discriminative yet robustly aligned space. In prior work, during the training of the alignment module of the EA models, for each anchor entity $e$, the set of negative samples is sampled uniformly from the data without accounting for their informativeness with regard to the corresponding anchor entity. Studies show that informative negative samples can guide the training phase, as they are intuitively the ones that are mapped in close proximity, but instead should be far apart (Robinson et al., 2021; Chen et al., 2020). The proposed model consists of three variants of contrastive loss: (i) contrastive loss with top-$k$ hard negatives, (ii) contrastive loss with aggregated top-$k$ hard negatives, and (iii) In-batch NT-Xent loss, where all the negatives are used. Let $f_\theta(x) \in \mathbb{R}^d$ be the encoder output, i.e., the entity embeddings. Before computing similarities, the embeddings of the source and target entities are normalised: $z_i^s = \frac{f_\theta(s_i)}{||f_\theta(s_i)||}, z_i^t = \frac{f_\theta(t_i)}{||f_\theta(t_i)||}$.

**Contrastive loss with top-$k$ hard negatives** The embedding of the source entity is considered as the query, and the set of target embeddings in the batch as keys. This loss selects top-$k$ hardest negatives for each query (source entity) and employs a softmax over the positive and $k$ negatives. Let the similarity matrix be,

$$A_{i,j} = z_i^{sT} \cdot z_j^t, \quad i \neq j$$

To ensure that the positive target is not selected as a negative, the diagonal is masked by setting its entries to $-\infty$. For each row $i$ in $A_{i,j}$, we compute the set $N_i$ of $k$ indices with the largest value of $A_{i,j}$, where $j \neq i$. The top-$k$ hardest negatives are selected $\mathcal{N}_i = \text{TopK}(A_{i,j} \mid j \neq i)$, where $N_i = \{j_1, ..., j_k\}$. Then we compute the logits

$$\text{L}_i = \frac{1}{\tau}[A_{i,i}, A_{i,j_1}, ...., A_{i,j_k}] \in \mathbb{R}^{(1+k)},$$

where $\tau$ is the temperature. Next, the per-sample top-$k$ cross-entropy loss is obtained as

$$\ell_{i,j}^{TopK} = -\log \frac{\exp(\text{sim}(z_i^{sT}, z_i^t)/\tau)}{\exp(\text{sim}(z_i^{sT}, z_i^t)/\tau) + \sum_{j \in \mathcal{N}_i} \exp(\text{sim}(z_i^{sT}, z_j^t)/\tau)}$$

and the corresponding batch loss is $\frac{1}{m} \sum_{i,j=1}^{m} \ell_{i,j}, i \neq j$, where $m$ is the batch size.

**Contrastive loss with aggregated top-$k$ hard negatives** In this loss function, for each source entity (query), after selecting the relevant top-$k$ hard negative samples as described above, we compute their mean similarities to obtain a single aggregated negative score per entity. This is mathematically given by

$$\text{neg}_i = \frac{1}{k} \sum_{j \in \mathcal{N}_i} A_{i,j}$$

where $\text{neg}_i$ is the aggregated negative score for source entity $i$, $k$ is the number of negatives, and $\mathcal{N}_i$ is the list of top-$k$ hardest negatives for the entity $i$.

**NT-Xent loss with in-batch negatives**  Our contrastive learning approach also explores a diverse sample of negatives for a given anchor entity using the NT-Xent loss with in-batch negatives (Chen et al., 2020). The goal of this loss is to maximise the similarity between positive pairs (true source entity – target entity pair) and minimise the similarity between negative pairs (incorrect source entity – target entity pair). Given a batch of $m$ normalised pairs of source and target entities $\{z_i^s, z_i^t\}_{i=1}^m$, a similarity matrix $A_{i,j}$ of logits is generated, where $A \in \mathbb{R}^{m \times m}$ with entries $A_{i,j} = \frac{z_i^{s^T} \cdot z_j^t}{\tau}, i \neq j$, where $\tau$ is the temperature. Here, all pairs of cosine similarity are scaled by a temperature value before computing the cross-entropy loss. For each query $i$ (source entity), the positive key (target entity) is $j = i$, and for each source entity, the cross-entropy loss is given by

$$\ell_{i,j}^{\text{NT-Xent}} = -\log \frac{\exp(A_{i,i})}{\sum_{j=1}^m \exp(A_{i,j})} = -A_{i,i} + \log \left( \sum_{j=1}^m A_{i,j} \right),$$

where $m$ is the batch size, and the batch loss is given by $\frac{1}{m} \sum_{i,j=1}^m \ell_{i,j}, i \neq j$.

Here, the embedding of the source entity is considered for each input, and we then compute the cosine similarity between every pair of target entity vectors. Since each entity in the source KG is similar to exactly one other entity in the target KG (positive pair), it can be considered as a classification task where similarity scores correspond to the probability distribution, in which the correct targets approach 1.0 and the rest will be close to 0.0.

**Why Hard Negatives Matter?**  For the NT-Xent loss, the partial derivative of $\ell_i^{\text{NT-Xent}}$ with respect to $\text{sim}(z_i^s, z_{n^-}^t)$ for a specific negative $n^-$ is proportional to a softmax weight

$$\frac{\partial \ell_{i,j}^{\text{NT-Xent}}}{\partial \text{sim}(z_i^s, z_{n^-}^t)} \, \alpha \, \frac{\exp(\text{sim}(z_i^s, z_{n^-}^t)/\tau)}{\sum_{i \neq a} \exp(\text{sim}(z_i^s, z_{a^-}^t)/\tau)}$$

Therefore, negatives with larger similarity (hard negatives) will have larger gradients; however, the collective impact of many easy negatives might overpower the impact of a few hard negatives in the training. Therefore, restricting the denominator to only the hard negatives $\mathcal{N}_i$, as demonstrated in $\ell_{i,j}^{\text{TopK}}$, will explicitly increase the influence of the negatives with large similarity by concentrating the gradients.

## 3 Experiments and Results

### 3.1 Datasets

**Benchmark Datasets**  The most commonly used cross-lingual EA benchmark datasets are DBP15K (Sun et al., 2017), SRPRS (Guo et al., 2019), and DBP5L (Chakrabarti et al., 2022) (detailed statistics are provided in Table 1). Although they are sampled from the real-world KGs DBpedia, Wikidata and YAGO, their entity distributions do not reflect the actual distributions in the original KGs. These real-world KGs typically follow a power-law degree distribution with more than 50% of the entities having degrees less than 3. In contrast, the DBP15K dataset, which contains aligned entities from three language pairs ZH–EN, JA–EN, and FR–EN, within which 21% to 43% of the entities have a degree less than or equal to 3. Similarly, in DBP5L, 33–56% of entities have a degree less than or equal to 3. This over-representation of the connectivity of the entities in the DBP15k dataset is rectified in SRPRS dataset, in which only 77–85% of entities have degrees less than or equal to 3. The detailed degree distribution of the entities in these benchmarks datasets is shown in Figure 6 and in Table 8 in Appendix A.1.

**LT-EA-25k dataset**  Although the SRPRS dataset includes low-degree entities, it is restricted to only two languages – German and French – and their corresponding alignment to English, limiting its usage in the evaluation of extensive cross-lingual EA. Moreover, the connectivity and completeness of the KGs vary from language to language. For instance, English or German DBpedia is denser

Table 1: Statistics of the benchmark datasets DBP15k, SRPRS, and DBP5L

| Dataset | Lang. | Entities | Relations | Rel. Triples | Attr. Triples |
|---------|-------|----------|-----------|--------------|---------------|
| DBP15K | ZH | 19,388 | 1,701 | 70,414 | 379,684 |
| (ZH-EN) | EN | 19,572 | 1,323 | 95,142 | 567,755 |
| DBP15K | JA | 19,814 | 1,299 | 77,241 | 354,619 |
| (JA-EN) | EN | 19,780 | 1,153 | 93,484 | 497,230 |
| DBP15K | FR | 19,661 | 903 | 105,998 | 528,665 |
| (FR-EN) | EN | 19,993 | 1,208 | 115,722 | 576,543 |
| SRPRS | EN | 15,000 | 222 | 38,363 | 62,715 |
| (EN-DE) | DE | 15,000 | 120 | 37,377 | 142,506 |
| SRPRS | EN | 15,000 | 221 | 36,508 | 70,750 |
| (EN-FR) | FR | 15,000 | 177 | 33,532 | 56,344 |
| | EN | 13,996 | 831 | 80,167 | – |
| | FR | 13,176 | 178 | 49,015 | – |
| DBP-5L | ES | 12,382 | 144 | 54,066 | – |
| | JA | 11,805 | 128 | 28,774 | – |
| | EL | 5,231 | 111 | 13,839 | – |

compared to Hindi or Arabic DBpedia. Therefore, a dataset with more languages would also change the proportion of the degree distribution of the entities. Also, none of these benchmark datasets consider *dangling entities* in the KGs, i.e., entities with no connections to other entities.

To address these limitations, we introduce a new cross-lingual long-tailed entity alignment dataset **LT-EA-25k** comprising seven languages – Arabic, German, Hindi, Italian, Japanese, Portuguese, and Russian and their corresponding alignment with English curated from DBpedia. The proportion of entities with a degree $\leq 3$ varies between 42% and 81% depending on the language, reflecting the sparsity and heterogeneity of multilingual KGs. To curate the proposed dataset, we follow these steps for each language-specific KG:

- We perform degree profiling for all the entities – i.e., we compute the outdegree, indegree, and the total degree of each entity.

- We divide the entities into 6 buckets based on their degree values, i.e $\{1, 2, 3, 4, 5, >5\}$.

- Next, we select entities from each bucket using a stratified random sampling to ensure a true representative distribution of the entities in each language edition of DBpedia, and thereby avoiding high connectivity bias as observed in the DBP15k and DBP5L datasets.

- The corresponding aligned entities from English DBpedia are extracted by exploiting the owl:sameAs links using SPARQL queries.

We also include smaller language editions of DBpedia, namely Hindi. Therefore, LT-EA-25k contains only 4,296 aligned entities between Hindi and English while maintaining the degree distribution of the entities. The Arabic edition of DBpedia is also very sparse, as evidenced in our dataset, which contains 11,617 entities with degree 1. We also incorporated ∼1,400 dangling entities from the Arabic DBpedia. A detailed statistics of our dataset is provided in Table 2 and the degree distribution in Figure 1 and in Table 9 of Appendix A.1. Unlike prior works, we also include medium and low-resource language KGs in our dataset, instead of merely high-resource language-specific KGs that are well-connected. Overall, we provide a robust dataset mirroring the real-world KGs by considering multilinguality, heterogeneity, and sparsity.

## 3.2 EXPERIMENTAL SETUP

**Implementation**    In our experiments, we use the mE5-base [2] encoder (Wang et al., 2024) to produce normalised text embeddings for entity names. The model is initialised with XLM-RoBERTa-base model, has 12 layers, and the embedding dimensionality is 768. It supports ∼100 languages, which is crucial for cross lingual EA. For robust training of **ContrastEA**, we perform a hyperparameter search for the following values. For the learning rate: {0.00001, 0.00002, 0.00003, 0.00004,

---

[2]https://huggingface.co/intfloat/multilingual-e5-base

Table 2: LT-EA-25k Dataset Statistics

| Lang. Pairs | Aligned Entities | Entities with Desc | Triples Source Lang. | Triples Target Lang. | Attr. Source Lang. | Attr. Target Lang. |
|---|---|---|---|---|---|---|
| AR–EN | 25,000 | 24,712 | 66,613 | 166,740 | 29,293 | 91,672 |
| DE–EN | 25,000 | 24,763 | 107,418 | 117,647 | 146,926 | 94,519 |
| HI–EN | 4,296 | 4,252 | 30,921 | 31,425 | 4,960 | 8,600 |
| IT–EN | 25,000 | 19,503 | 106,589 | 126,566 | 87,206 | 83,034 |
| JA–EN | 25,000 | 24,491 | 115,373 | 130,072 | 56,408 | 69,939 |
| PT–EN | 25,000 | 24,155 | 111,742 | 111,902 | 75,660 | 101,007 |
| RU–EN | 25,000 | 24,766 | 101,047 | 132,867 | 60,343 | 85,219 |

Figure 1: Degree Distribution of the proposed LT-EA-25k dataset

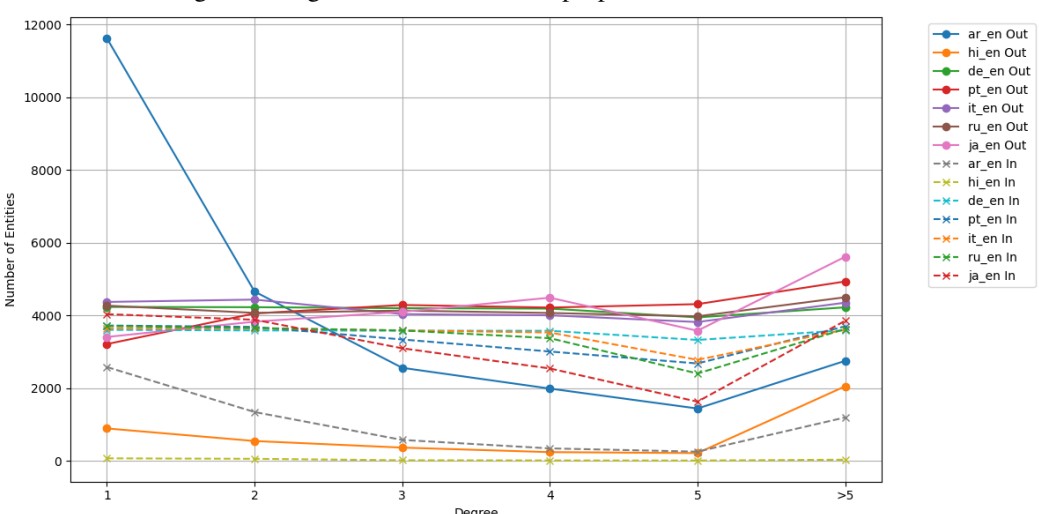

0.00005, 0.0001, 0.0002, 0.0003, 0.0004, 0.0005, 0.001, 0.002, 0.003, 0.004, 0.005}, batch size: {16, 32, 64}, epochs: {3, 5, 10, 15, 20}, warmup ratio: {0.1, 0.5, 0.8}, top-k (for hard negatives in contrastive learning): {3, 5, 10, 15}, weight decay: {0.1, 0.01, 0.001, 0.0001}, and temperature:{0.1, 0.01, 0.001, 0.0001}. The selected settings are: learning rate: 2e-5, batch size: 32, epochs: 5, warmup ratio: 0.1, top-k: 5, weight decay: 0.01, and temperature: 0.1. We also fix the random seed at 42 for reproducibility. Topk = 5 indicates that during training for each source entity, the most similar 5 target entities are selected as negative samples. The analysis with different k and batch size is provided in the Appendix A We use 30% of the aligned entities as the training set and the remaining 70% as the test set. Following the convention, we report results for Hits@1 and Hits@10 for all three benchmark datasets – DBP15k, SRPRS, and DBP5L, as well as our curated dataset LT-EA-25k. The experiments have been carried out in NVIDIA Tesla A100 GPU with 80GB memory.

### 3.3 RESULTS

**Evalution on Benchmark Datasets** As shown in Tables 3 and 4, our proposed model ContrastEA achieves SOTA results in both SRPRS and DBP15k datasets for both Hits@1 (exact match) and Hits@10. On the SRPRS dataset, for Hits@1 ContrastEA achieves a steep improvement of 20 percentage points for EN–FR language pair, and ∼10 percentage points for EN–DE over the previous best method, DAT. The DAT Zeng et al. (2020) model focuses on the alignment of long-tailed entities and uses entity names as a feature, together with the neighbourhood information of entities. Since Hits@1 is equivalent to precision, this result highlights that most of the entities are correctly aligned at the highest rank by the ContrastEA model.

On the other hand, Hits@10 reaches 0.978 (EN–FR) and 0.99 (EN–DE), indicating that the majority of the correct alignments are captured within the top 10 candidates, demonstrating robust recall. Al-

Table 3: Results on SRPRS dataset

| Methods | EN-FR | | EN-DE | |
|---|---|---|---|---|
| | Hits@1 | Hits@10 | Hits@1 | Hits@10 |
| MTransE | 25.1 ↓70.9 | 55.1 ↓42.7 | 31.2 ↓65.8 | 58.6 ↓40.4 |
| IPTransE | 25.5 ↓70.5 | 55.7 ↓42.1 | 31.3 ↓65.7 | 59.2 ↓39.8 |
| BootEA | 31.3 ↓64.7 | 62.9 ↓34.9 | 44.2 ↓52.8 | 70.1 ↓28.9 |
| RSNs | 34.8 ↓61.2 | 63.7 ↓34.1 | 49.7 ↓47.3 | 73.3 ↓25.7 |
| MuGNN | 13.1 ↓82.9 | 34.2 ↓63.6 | 24.5 ↓72.5 | 43.1 ↓55.9 |
| KECG | 29.8 ↓66.2 | 61.6 ↓36.2 | 44.4 ↓52.6 | 70.7 ↓28.3 |
| TransEdge | 40.0 ↓56.0 | 67.5 ↓30.3 | 55.6 ↓41.4 | 75.3 ↓23.7 |
| GCN | 15.5 ↓80.5 | 34.5 ↓63.3 | 25.3 ↓71.7 | 46.4 ↓52.6 |
| JAPE | 25.6 ↓70.4 | 56.2 ↓41.6 | 32.0 ↓65.0 | 59.9 ↓39.1 |
| RDGCN | 67.5 ↓28.5 | 76.9 ↓20.9 | 78.3 ↓18.7 | 88.4 ↓10.6 |
| HGCN | 67.0 ↓29.0 | 77.0 ↓20.8 | 76.3 ↓20.7 | 86.0 ↓13.0 |
| GM-Align | 62.7 ↓33.3 | – | 67.7 ↓29.3 | – |
| DAT | 75.8 ↓20.2 | 89.9 ↓7.9 | 87.6 ↓9.4 | 95.0 ↓4.0 |
| ContrastEA | **96.0** | **97.8** | **97.0** | **99.0** |

Table 4: Results on DBP-15k dataset.

| Methods | ZH-EN | | JA-EN | | FR-EN | |
|---|---|---|---|---|---|---|
| | H@1 | H@10 | H@1 | H@10 | H@1 | H@10 |
| MTransE | 30.8 ↓29.4 | 61.4 ↓16.4 | 27.9 ↓49.5 | 57.5 ↓30.4 | 24.4 ↓71.6 | 55.6 ↓43.6 |
| IPTransE | 40.6 ↓19.6 | 73.5 ↓4.3 | 36.7 ↓40.7 | 69.3 ↓18.6 | 33.3 ↓62.7 | 68.5 ↓30.7 |
| JAPE | 41.2 ↓19.0 | 74.5 ↓3.3 | 36.3 ↓41.1 | 68.5 ↓19.4 | 32.4 ↓63.6 | 66.7 ↓32.5 |
| AlignE | 47.2 ↓13.0 | 79.2 ↑1.4 | 44.8 ↓32.6 | 78.9 ↓9.0 | 48.1 ↓47.9 | 82.4 ↓16.8 |
| GCN-Align | 41.3 ↓18.9 | 74.4 ↓3.4 | 39.9 ↓37.5 | 74.5 ↓13.4 | 37.3 ↓58.7 | 74.5 ↓24.7 |
| SEA | 42.4 ↓17.8 | 79.6 ↑1.8 | 38.5 ↓38.9 | 78.3 ↓9.6 | 40.0 ↓56.0 | 79.7 ↓19.5 |
| RSN | 50.8 ↓9.4 | 74.5 ↓3.3 | 50.7 ↓26.7 | 73.7 ↓14.2 | 51.6 ↓44.4 | 76.8 ↓22.4 |
| MuGCN | 49.4 ↓10.8 | 84.4 ↑6.6 | 50.1 ↓27.3 | 85.7 ↓2.2 | 49.5 ↓46.5 | 87.0 ↓12.2 |
| GCN | 48.7 ↓11.5 | 79.0 ↑1.2 | 50.7 ↓26.7 | 80.5 ↓7.4 | 50.8 ↓45.2 | 80.8 ↓18.4 |
| GAT | 41.8 ↓18.4 | 66.7 ↓11.1 | 44.6 ↓32.8 | 69.5 ↓18.4 | 44.2 ↓51.8 | 73.1 ↓26.1 |
| R-GCN | 46.3 ↓13.9 | 73.4 ↓4.4 | 47.1 ↓30.3 | 75.4 ↓12.5 | 46.9 ↓49.1 | 75.8 ↓23.4 |
| AliNet | 53.9 ↓6.3 | 82.6 ↑4.8 | 54.9 ↓22.5 | 83.1 ↓4.8 | 55.2 ↓40.8 | 85.2 ↓14.0 |
| BootEA | 62.9 ↑2.7 | 84.7 ↑6.9 | 62.3 ↓15.1 | 85.4 ↓2.5 | 65.3 ↓30.7 | 87.4 ↓11.8 |
| Dual-AMN | **73.1** ↑12.9 | **92.4** ↑14.6 | 72.6 ↓4.8 | **92.7** ↑4.8 | 75.6 ↓20.4 | 94.8 ↓4.4 |
| ContrastEA | 60.2 | 77.8 | **77.4** | 87.9 | **96.0** | **99.2** |

Table 5: Results on DBP5L dataset.

| LangPair | AlignKGC | RAGA | RNM | RDGCN | ContrastEA |
|---|---|---|---|---|---|
| EL-EN | 83.8 ↓6.2 | 75.8 ↓14.2 | 74.9 ↓15.1 | 71.3 ↓17.7 | **89.0** |
| EL-ES | 83.3 ↓4.7 | 79.8 ↓8.2 | 79.4 ↓8.6 | 74.7 ↓13.3 | **88.0** |
| EL-FR | 82.1 ↓2.9 | 69.1 ↓15.9 | 72.4 ↓12.6 | 72.7 ↓12.3 | **85.0** |
| EL-JA | 74.8 ↓4.2 | 64.4 ↓14.6 | 68.3 ↓10.7 | 64.4 ↓14.6 | **79.0** |
| JA-EN | 76.5 ↓9.1 | 59.1 ↓26.5 | 64.5 ↓21.1 | 58.2 ↓27.4 | **85.6** |
| JA-ES | 74.3 ↓8.4 | 58.3 ↓24.4 | 65.0 ↓17.7 | 60.0 ↓22.7 | **82.7** |
| JA-FR | 73.9 ↓7.4 | 64.5 ↓16.8 | 70.6 ↓10.7 | 60.2 ↓21.1 | **81.3** |
| ES-FR | 89.5 ↓4.4 | 80.9 ↓13.0 | 84.9 ↓9.0 | 87.1 ↓2.9 | **93.9** |
| ES-EN | 93.3 ↓4.4 | 85.7 ↓12.0 | 88.0 ↓9.7 | 87.8 ↓9.9 | **97.7** |
| EN-FR | 90.5 ↓4.0 | 77.0 ↓17.5 | 81.2 ↓13.3 | 83.2 ↓11.3 | **94.5** |

though all baseline models use the structural and relational information of the entities, our empirical results demonstrate that the entity representations obtained from the pre-trained LM coupled with the contrastive objectives are more effective. Similarly, in the DBP15k dataset, the large improvement in the FR-EN language pair shows that our proposed model is better suited for cross-lingual alignment of closely related languages, confirming the effectiveness of the entity embeddings. For both JA-EN and ZH-EN, ContrastEA outperforms the second best model AliNet, by 22.5 and 6 percentage points, respectively, indicating that the model can overcome linguistic barriers such as different scripts. For Hits@10, ContrastEA achieves comparable results for ZH-EN language pair

Table 6: Results for the LT-EA-25k dataset

| Language Pair | Features | ContrastEA | | | DAT | |
|---|---|---|---|---|---|---|
| | | Hits@1 | Hits@3 | Hits@10 | Hits@1 | DAT Hits@10 |
| **AR-EN** | Name | 81.42 | 88.39 | 92.67 | 1.44 | 4.78 |
| | Desc | 58.17 | 63.19 | 66.81 | | |
| | Name+Desc | 83.33 | 89.10 | 93.14 | | |
| **HI-EN** | Name | 86.17 | 92.82 | 95.55 | 1.36 | 6.56 |
| | Desc | 87.23 | 94.59 | 96.55 | | |
| | Name+Desc | 88.20 | 95.20 | 96.80 | | |
| **IT-EN** | Name | 95.72 | 96.91 | 97.67 | 0.75 | 3.93 |
| | Desc | 86.14 | 89.35 | 92.53 | | |
| | Name+Desc | 96.14 | 97.05 | 97.91 | | |
| **RU-EN** | Name | 90.12 | 94.10 | 96.09 | 0.56 | 2.68 |
| | Desc | 87.54 | 92.57 | 95.83 | | |
| | Name+Desc | 90.74 | 94.87 | 96.88 | | |
| **JA-EN** | Name | 53.33 | 59.73 | 65.95 | 0.97 | 3.40 |
| | Desc | 83.60 | 89.22 | 92.79 | | |
| | Name+Desc | 85.23 | 92.56 | 93.87 | | |
| **PT-EN** | Name | 96.11 | 97.84 | 98.38 | 1.05 | 4.60 |
| | Desc | 94.74 | 97.06 | 98.15 | | |
| | Name+Desc | 96.77 | 97.90 | 98.72 | | |
| **DE-EN** | Name | 97.31 | 98.49 | 99.05 | 0.59 | 3.39 |
| | Desc | 93.74 | 96.10 | 97.46 | | |
| | Name+Desc | 97.90 | 98.72 | 99.23 | | |

Figure 2: Results of ContrastEA on LT-EA-25k dataset w.r.t. degree distribution

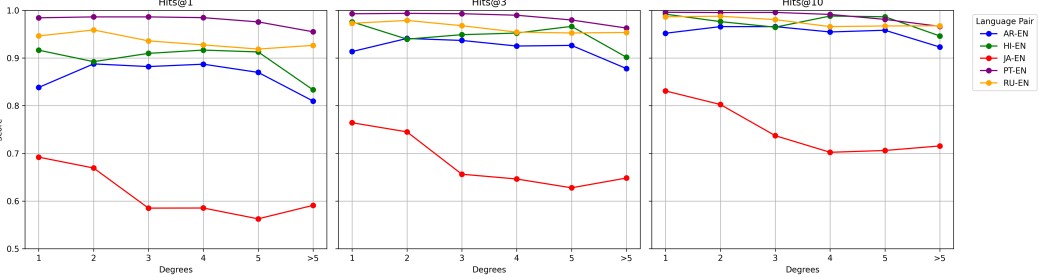

while outperforming for JA-EN and FR-EN. The GCN based models achieve high performance for Hits@10 but not for Hits@1, indicating the structural information of entities is insufficient to distinguish between similar deceptive entities, while contrastive learning ensures language separation.

Table 5 shows that ContrastEA also achieves the SOTA for all the language pairs in DBP5L dataset. While RDGCN (Wu et al., 2019a) leverages the relational information, AlignKGC (Chakrabarti et al., 2022), RAGA (Zhu et al., 2021a), and RNM (Zhu et al., 2021b) includes textual information. This dataset consists of several European languages, and the results confirm that ContrastEA is capable of capturing the fine-grained semantic characteristics of closely-related languages. The model also achieves the best results for the distant language pairs, namely, JA-FR, JA-ES and JA-EN. These language pairs are most challenging due to their linguistic differences as well as the scripts. ContrastEA shows an improvement of 5-9 percentage points over the existing best baseline model, AlignKGC. Therefore, it can be concluded that ContrastEA is robust across different language pairs with linguistic variations. Intuitively, it can be presumed that the KG embedding based or GNN-based EA models rely mostly on the structural information, precisely the local neighbourhood, and therefore fail to capture the true representations when entities are sparse or noisy. Furthermore, the topk hard negative contrastive learning plays a vital role in separating the true pairs from the deceptive ones. Further results are available in Appendix A.2.

**Evaluation on LT-EA-25k dataset**   We conducted experiments using entity names, descriptions, and their combination. The results show that combining both features yields the strongest performance. We also performed experiments with zero-shot LLMs (Qwen 32B and DeepSeek-R1-70bn, results in Table 11.

**Performance using Entity Names** The results in Table 6 highlight a clear performance gap between ContrastEA and DAT across almost all language pairs. DAT has relatively very low, with Hits@1 $\leq$ 1.44 for all language pairs, suggesting that it has trouble successfully aligning entities on this dataset. We trained our dataset on the DAT [3] model as it is focused on long-tailed entities, and also, as shown in Table 3, it achieves the second-best performance after ContrastEA. As mentioned in Section 3.1, the proposed LT-EA-25k dataset includes a large proportion of long-tailed entities with connectivity less than 3. Since the existing models, including DAT, rely heavily on the neighbourhood connection, the models are incapable of handling sparse KG; therefore do not produce entity embeddings that are well represented. Moreover, the language pair AR-EN is particularly challenging as it contains 1400 dangling entities, for which there is no neighbourhood information. Despite this, ContrastEA achieves a high Hits@1 (81.42%), whereas DAT achieves a low Hits@1. The hardest language pair for ContrastEA appears to be JA-EN, which is due to the linguistic differences, besides a larger number of long-tailed entities. Similar observation has also been made in general for JA-EN, also in DBP15K datasets, as illustrated in Table 4. However, the alignment of the European languages IT, PT, DE, and RU with English achieves stronger results, indicating higher language similarities and effective separation of dissimilar entities from the similar ones. Another advantage of ContrastEA is that no training is required to generate the entity embeddings.

**Performance using Entity Descriptions**  We conducted additional experiments on our LT-EA-25k dataset using 1-sentence entity descriptions wherever available to analyse if they are useful for long-tailed entities in low-resource languages. The results are presented in the Table 6, revealing several important trends. We observe the most notable gain for **Japanese**, where **Hits@1** jumps from $53.33 \rightarrow 83.60$. This supports the hypothesis that Japanese entities often have short or transliterated surface forms that provide very weak semantic signals, whereas even brief descriptions supply rich contextual cues. For e.g. the entity 白い暴動 (アルバム) translates to *White Riot (album)* and our prediction model with entity names mapped it to the *White Riot* entity from English DBpedia, but the correct corresponding English entity is *The Clash (album)*. But, when sentence description is used instead of the entity name, it could correctly map it to the English entity *The Clash (album)*. Therefore, descriptions substantially improve alignment for low-resource languages or script-divergent languages. It also shows mild improvements in Hindi. The entity type information embedded in the entity names proves to be beneficial for entity alignment. This can be concluded from the Table 6, for e.g., Arabic shows a significant degradation ($81.42 \rightarrow 58.17$ in Hits@1). We inspected this phenomenon closely and found that for entities *Star Wars (film)* got correctly aligned to their corresponding target entity when only names are used. When switching to descriptions alone, this strong built-in disambiguation signal is lost, leading to poorer performance. In such cases, the name itself is more informative than descriptions. Motivated by these findings, we evaluated our dataset using combining both names and descriptions and the results are provided in the Table 6.

**Performance on Degree distribution**   We report the ablation studies of ContrastEA on the degree distribution of the entities in the LT-EA-25k dataset, as shown in Figure 2 and in Table 7. We reported results on Hits@1, Hits@3 and Hits@10 across degrees (1, 2, 3, 4, 5, >5). The model achieves remarkable performance on Hits@1 with Degree = 1,2, with HI and PT achieving the highest values 0.9165 (for Deg=1), and 0.8924 (for Deg=2) and 0.9844 (for Deg=1), 0.9863 (for Deg=2), respectively. These values show that the model is robust with sparsely connected entities and also agnostic towards their linguistic differences. We observe that with Hits@1 and Hits@3, the performance of the model drops a little across most of the languages, with entities having degrees more than 5. However, this is an aggregated performance of the model for all entities. Overall, high values of Hits@1 and Hits@10 across different degrees indicate that ContrastEA is consistent with high precision and recall. Since a major proportion of the curated LT-EA-25k dataset is sparse, replicating the entity distribution of real-world KGs, we infer from the ablation studies that ContrastEA is capable of learning ambiguous (learning deceptive entities in TopK loss) entity patterns in sparse settings. As our method requires no training to generate representations and does not rely on neighborhood or relational triples, it scales effectively to large multilingual KGs—covering both

---

[3] https://github.com/DexterZeng/DAT

long-tailed, low-resource entities and popular, high-resource ones. For all languages, the Hits@k metrics and Accuracy are higher for entities with degrees 1–3 compared to entities with degrees 4, 5, and $> 5$. This indicates that entities with lower connectivity in the KG (degree 1–3) are easier to align, likely because they are less entangled with other entities and have fewer potential hard negatives. However, popular entities containing more information tend to be comparatively ambiguous. Japanese suffers the most with high-degree entities due to short/transliterated surface forms and low-resource characteristics, while other languages such as PT, RU, HI maintain high performance, showing the robustness of the model with relevant disambiguating information.

## 4 RELATED WORK

Recent research on entity alignment (EA) broadly falls into two categories: semantic matching-based models and graph neural network (GNN)-based models (Zeng et al., 2021; Sun et al.). Semantic matching-based models aim to embed entities in a low-dimensional space such that semantically similar entities are close together. Early approaches like MTransE (Chen et al., 2017), IP-TransE (Zhu et al., 2017), rely on translational embeddings (e.g., TransE) and linear transformations for alignment. Models such as JAPE (Sun et al., 2017) incorporate attributes or textual descriptions, semi-supervised or iterative frameworks like BootEA (Sun et al., 2018), to improve alignment accuracy. Recent methods, e.g., BERT-INT (Tang et al., 2021), leverage pre-trained language models to encode entity names and descriptions. GNN-based models exploit the graph structure of knowledge graphs, encoding nodes based on neighbour information to capture local and global structural patterns. GCN-Align (Xu et al., 2019) pioneered this direction, followed by models like RDGCN (Wu et al., 2019a), which include relational information. HMAN (Yang et al., 2019) further integrates entity attributes, textual descriptions, and pre-trained embeddings within GNN frameworks. Other models include AttrGNN (Liu et al., 2020), and REA (Pei et al., 2020), which collectively improve alignment by leveraging graph topology and semantics. Recent works (Zeng et al., 2020) have also used entity names as features together with structural and neighbourhood information, while Lambda Yin et al. (2024) uses the KEESA GNN encoder with selective aggregation and a dangling entity detector for noise removal with a contrastive loss. .

## 5 CONCLUSION

In this paper, we introduce a novel cross-lingual entity alignment method, ContrastEA, which achieves state-of-the-art across the three widely used benchmark datasets and our proposed LM-EA25k dataset. The experimental results demonstrate consistent and robust performance across entities with varying degrees - popular and under-represented entities in a KG, namely long-tailed, and dangling entities. By leveraging the multilingual LM, the model achieves high precision (Hits@1) across diverse language pairs with linguistic differences, including different scripts, word order, and typologically distant languages from different language families. Therefore, it can be concluded that our model does not overfit to the language-specific structures and thereby improves the generalizability of our approach, advancing the knowledge fusion on a broader spectrum. Our proposed dataset would also pave the way for future research in long-tailed and dangling entity alignment across different KGs.

In the future, we intend to focus on incorporating a short description of the entities in the model, which would help us disambiguate between highly correlated yet deceptive entities. As KGs are inherently sparse and are continuously evolving, our focus would be on multilingual KG completion with the help of cross-lingual EA.

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

## A  APPENDIX

Table 7: Aggregated performance by degree groups (1–3 vs. 4–>5) across languages

| Language | Degree | Hits@1 | Hits@3 | Hits@10 |
|---|---|---|---|---|
| AR-EN | Deg 0 | 61.82 | 64.35 | 68.70 |
| AR-EN | Deg 1–3 | 86.94 | 93.05 | 96.13 |
| AR-EN | Deg 4–>5 | 85.56 | 90.99 | 94.54 |
| HI-EN | Deg 1–3 | 90.62 | 95.48 | 97.77 |
| HI-EN | Deg 4–>5 | 88.76 | 94.02 | 97.37 |
| JA-EN | Deg 1–3 | 64.89 | 72.19 | 79.02 |
| JA-EN | Deg 4–>5 | 57.69 | 63.81 | 71.09 |
| PT-EN | Deg 1–3 | 98.57 | 99.32 | 99.57 |
| PT-EN | Deg 4–>5 | 97.19 | 97.75 | 97.97 |
| RU-EN | Deg 1–3 | 94.71 | 97.32 | 98.50 |
| RU-EN | Deg 4–>5 | 92.43 | 95.35 | 96.69 |

More details on the dataset:

- On the quality of owl:sameAs links: The owl:sameAs links in LT-EA-25K were directly extracted from DBpedia. These links are curated and maintained by the DBpedia Association, and therefore, we did not perform additional manual validation.

- On generalizability beyond DBpedia (e.g., Wikidata, YAGO): Our dataset specifically targets cross-lingual entity alignment across different DBpedia language editions. We did not include Wikidata because Wikidata is a single, language-independent KG where entities are uniquely identified (e.g., Q937 for Albert Einstein), and multilinguality appears only at the level of labels/surface forms. Consequently, standard cross-lingual entity alignment within the entities in Wikidata is not applicable in this context, since all language versions map to the same entity ID by design. However, the dataset construction methodology is not limited to DBpedia. It can be extended to YAGO by following the same pipeline.

- On experiments with other KGs: We did not evaluate ContrastEA on cross-KG benchmarks such as DBP-YAGO or DBP-Wikidata because existing benchmarks (e.g., SRPRS) use English-only surface forms, and therefore do not align with the multilingual focus of our work.

Figure 3: Hits@k vs Top-k Negatives

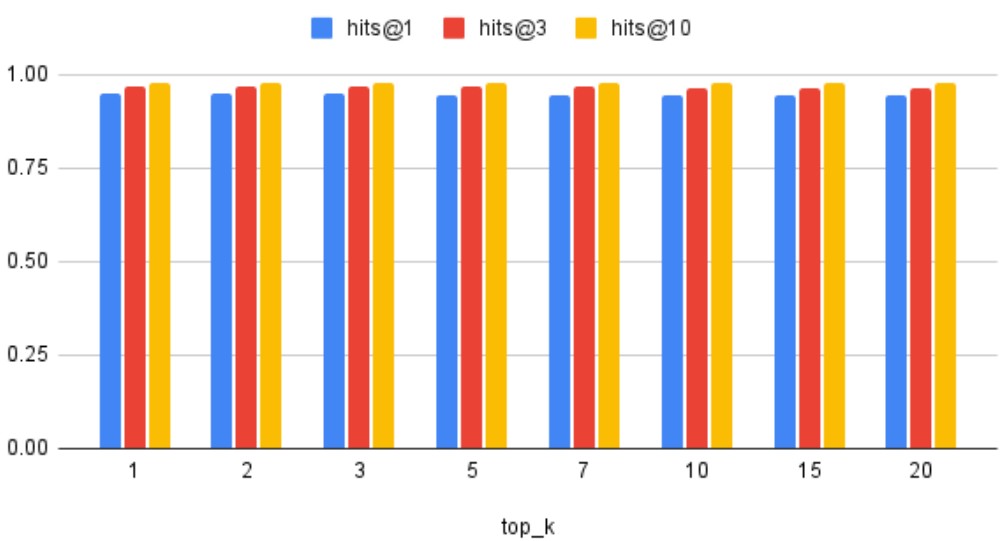

Figure 4: Hits@k vs batch size

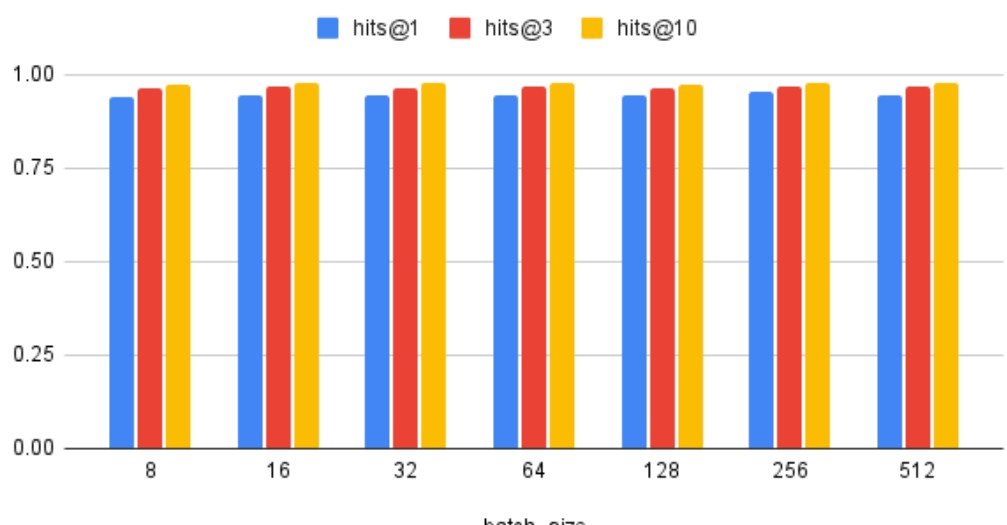

Hyperparameter Details

A.1  DEGREE DETAILS

A.2  RESULTS

Figure 5: Training loss vs batch size

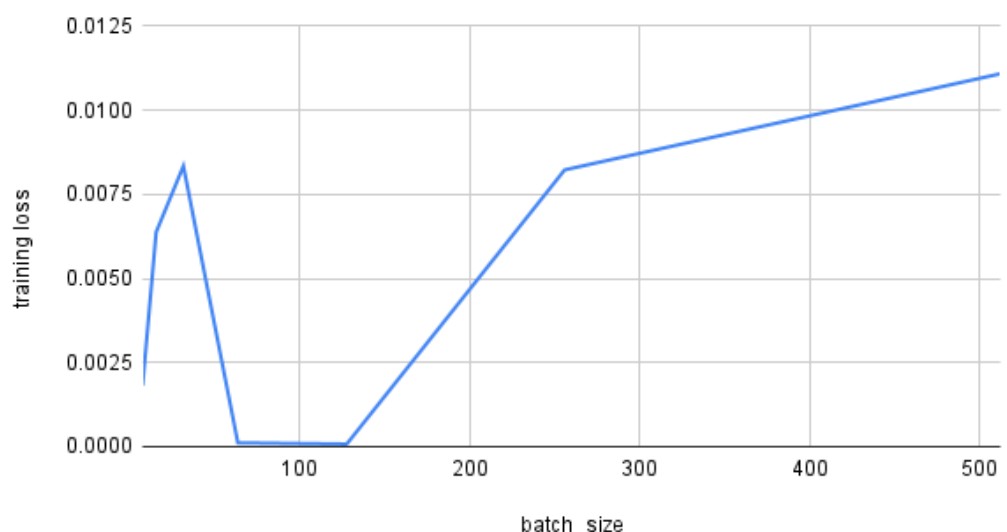

Table 8: Node outdegree distribution across different datasets and languages.

| Dataset / Lang | Outdeg=1 | Outdeg=2 | Outdeg=3 | Outdeg=4 | Outdeg=5 | Outdeg>5 |
|---|---|---|---|---|---|---|
| **SRPRS** | | | | | | |
| DBP | 3834 | 5527 | 2898 | 1172 | 646 | 923 |
| WD | 5155 | 5319 | 1765 | 784 | 493 | 1484 |
| **SRPRS** | | | | | | |
| DBP | 6679 | 4005 | 1816 | 988 | 618 | 894 |
| YG | 6687 | 4000 | 1480 | 832 | 536 | 1465 |
| **SRPRS** | | | | | | |
| EN | 5101 | 4285 | 2227 | 1384 | 818 | 1185 |
| DE | 6675 | 3670 | 1534 | 1063 | 636 | 1422 |
| **SRPRS** | | | | | | |
| EN | 5816 | 4464 | 1804 | 1102 | 696 | 1118 |
| FR | 5889 | 5229 | 1702 | 839 | 498 | 843 |
| **DBP-15K** | | | | | | |
| FR | 1126 | 1265 | 1105 | 1221 | 1359 | 8924 |
| EN | 967 | 1062 | 1053 | 1102 | 1228 | 9588 |
| **DBP-15K** | | | | | | |
| JA | 1385 | 1618 | 2016 | 2173 | 2457 | 5351 |
| EN | 1059 | 1363 | 1493 | 1718 | 2233 | 7134 |
| **DBP-15K** | | | | | | |
| ZH | 1704 | 2227 | 2499 | 2334 | 1776 | 4460 |
| EN | 954 | 1501 | 1742 | 1805 | 1760 | 7238 |
| **DBP5L Dataset** | | | | | | |
| EL | 836 | 695 | 450 | 376 | 350 | 830 |
| EN | 1097 | 1465 | 1524 | 1322 | 1175 | 5691 |
| ES | 1165 | 1216 | 889 | 867 | 695 | 4046 |
| FR | 841 | 1236 | 994 | 720 | 670 | 3834 |
| JA | 1181 | 871 | 604 | 510 | 374 | 2152 |

Figure 6: Degree distribution of all the benchmark datasets DBP15k (source-target), DBP5L (source-target), SRPRS (source-target)

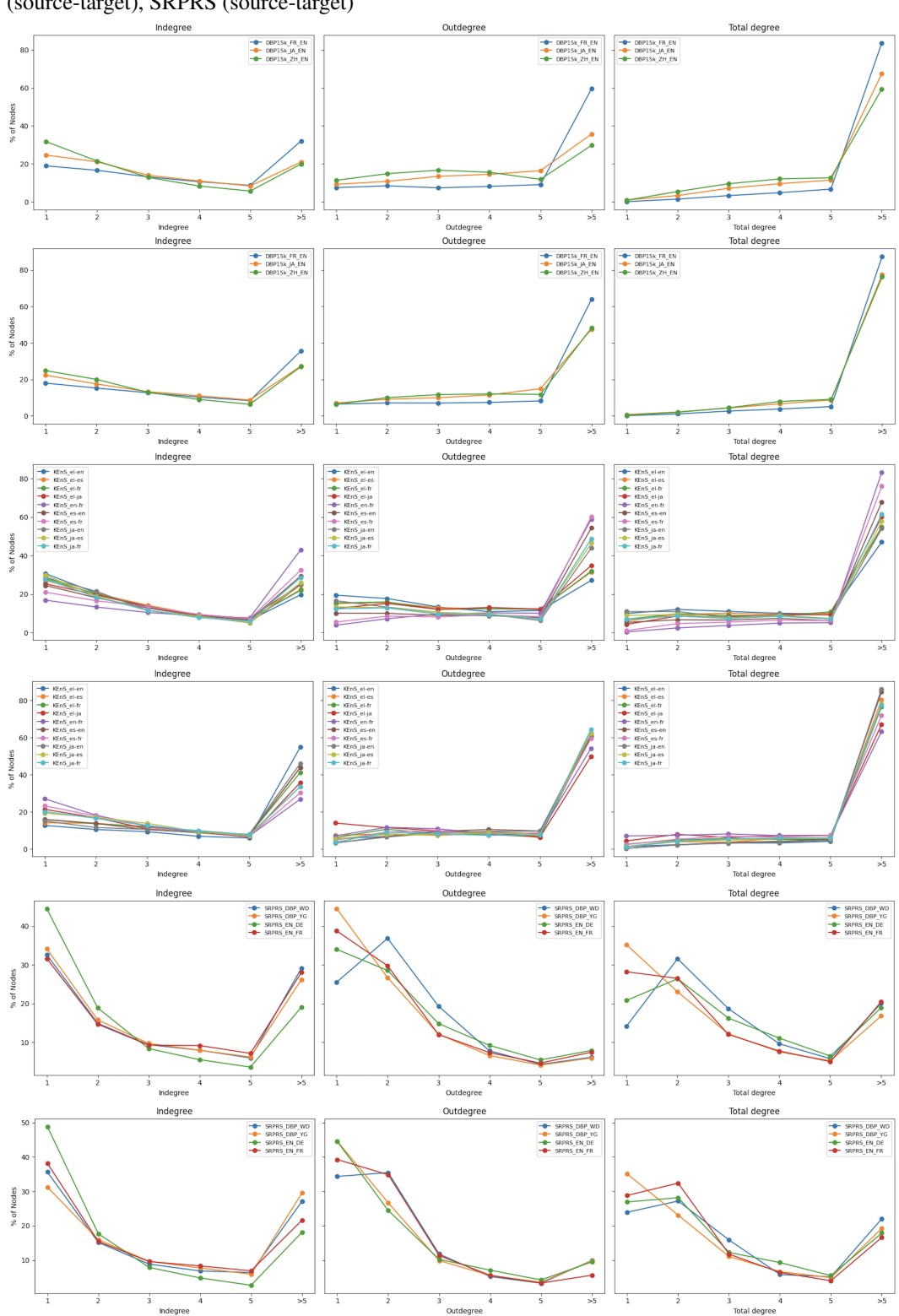

Table 9: Degree Distribution of the proposed LT-EA-25k dataset

| Lang Pairs | Outdegree Distribution | | | | | | Indegree Distribution | | | | | |
|---|---|---|---|---|---|---|---|---|---|---|---|---|
| | Deg=1 | Deg=2 | Deg=3 | Deg=4 | Deg=5 | Deg>5 | Deg=1 | Deg=2 | Deg=3 | Deg=4 | Deg=5 | Deg>5 |
| ar_en | 11,617 | 4,654 | 2,557 | 1,986 | 1,438 | 2,748 | 2,578 | 1,341 | 574 | 341 | 252 | 1,192 |
| hi_en | 890 | 544 | 363 | 239 | 212 | 2,048 | 68 | 55 | 15 | 10 | 8 | 31 |
| de_en | 4,231 | 4,224 | 4,197 | 4,185 | 3,943 | 4,220 | 3,601 | 3,585 | 3,577 | 3,575 | 3,323 | 3,590 |
| pt_en | 3,210 | 4,052 | 4,285 | 4,214 | 4,308 | 4,931 | 3,720 | 3,683 | 3,336 | 3,005 | 2,677 | 3,698 |
| it_en | 4,368 | 4,433 | 4,034 | 4,001 | 3,816 | 4,348 | 3,647 | 3,644 | 3,585 | 3,528 | 2,776 | 3,607 |
| ru_en | 4,267 | 4,067 | 4,131 | 4,065 | 3,973 | 4,497 | 3,708 | 3,652 | 3,585 | 3,372 | 2,401 | 3,623 |
| ja_en | 3,405 | 3,826 | 4,098 | 4,485 | 3,576 | 5,610 | 4,033 | 3,879 | 3,094 | 2,539 | 1,626 | 3,850 |

Table 10: ContrastEA results across different benchmarking datasets and language pairs.

| Dataset / Pair | Hits@1 | Hits@3 | Hits@10 | Accuracy |
|---|---|---|---|---|
| **DBP5L Dataset** | | | | |
| el_en | 0.8865 | 0.9451 | 0.9682 | 0.8865 |
| el_es | 0.8844 | 0.9396 | 0.9570 | 0.8844 |
| el_fr | 0.8508 | 0.9043 | 0.9416 | 0.8508 |
| el_ja | 0.7880 | 0.8520 | 0.8962 | 0.7880 |
| en_fr | 0.9459 | 0.9663 | 0.9742 | 0.9459 |
| es_en | 0.9778 | 0.9908 | 0.9956 | 0.9778 |
| es_fr | 0.9396 | 0.9618 | 0.9737 | 0.9396 |
| ja_en | 0.8569 | 0.9154 | 0.9500 | 0.8569 |
| ja_es | 0.8278 | 0.8907 | 0.9272 | 0.8278 |
| ja_fr | 0.8134 | 0.8707 | 0.9097 | 0.8134 |
| **SRPRS Dataset** | | | | |
| en_de | 0.9777 | 0.9869 | 0.9909 | 0.9777 |
| en_fr | 0.9612 | 0.9729 | 0.9786 | 0.9612 |
| **DBP-15K Dataset** | | | | |
| fr_en | 0.9607 | 0.9842 | 0.9921 | 0.9607 |
| ja_en | 0.7748 | 0.8398 | 0.8796 | 0.7748 |
| zh_en | 0.6017 | 0.6962 | 0.7786 | 0.6017 |

| Language Pair | Hits@1 |
|---|---|
| DE-EN | 19.2 |
| HI-EN | 12.1 |
| JA-EN | 10.4 |
| AR-EN | 11.8 |

Table 11: Hits@1 results for different language pairs on DeepSeek-R1-70bn

Table 12: Results on DBP15k dataset with mE5 + MUSE.

| method | src_lang | tgt_lang | train_size | Hits@1 | Hits@5 | Hits@10 | Hits@1 | Hits@5 | Hits@10 |
|--------|----------|----------|------------|--------|--------|---------|--------|--------|---------|
| unsupervised | fr | en | 30 | 95.183 | 98.502 | 98.941 | 97.243 | 99.370 | 99.609 |
| unsupervised | fr | en | 50 | 94.594 | 98.371 | 98.812 | 96.916 | 99.292 | 99.559 |
| unsupervised | fr | en | 70 | 93.486 | 98.155 | 98.644 | 96.243 | 99.289 | 99.511 |
| unsupervised | fr | en | 90 | 91.933 | 98.067 | 98.467 | 95.000 | 99.333 | 99.600 |
| unsupervised | ja | en | 30 | 70.309 | 81.840 | 84.564 | 79.365 | 87.960 | 90.071 |
| unsupervised | ja | en | 50 | 67.997 | 80.349 | 83.159 | 77.944 | 87.016 | 89.194 |
| unsupervised | ja | en | 70 | 65.194 | 78.583 | 81.431 | 75.847 | 85.580 | 87.755 |
| unsupervised | ja | en | 90 | 58.883 | 73.284 | 76.447 | 70.390 | 82.100 | 84.455 |
| unsupervised | zh | en | 30 | 51.129 | 67.151 | 72.020 | 64.999 | 78.592 | 82.456 |
| unsupervised | zh | en | 50 | 48.265 | 65.054 | 70.159 | 63.071 | 77.288 | 81.469 |
| unsupervised | zh | en | 70 | 45.767 | 63.301 | 68.684 | 61.269 | 76.368 | 80.612 |
| unsupervised | zh | en | 90 | 45.839 | 62.886 | 67.718 | 61.745 | 75.168 | 80.537 |
| supervised | fr | en | 30 | 94.868 | 98.388 | 98.884 | 97.310 | 99.351 | 99.599 |
| supervised | fr | en | 50 | 94.180 | 98.211 | 98.745 | 97.010 | 99.279 | 99.573 |
| supervised | fr | en | 70 | 92.752 | 97.977 | 98.577 | 96.398 | 99.266 | 99.533 |
| supervised | fr | en | 90 | 90.600 | 97.800 | 98.467 | 95.133 | 99.267 | 99.533 |
| supervised | ja | en | 30 | 68.803 | 80.190 | 83.193 | 79.672 | 87.903 | 90.061 |
| supervised | ja | en | 50 | 65.067 | 77.944 | 81.196 | 77.769 | 86.788 | 89.073 |
| supervised | ja | en | 70 | 59.206 | 74.322 | 78.157 | 75.196 | 85.064 | 87.464 |
| supervised | ja | en | 90 | 51.884 | 67.362 | 71.669 | 68.977 | 80.686 | 83.917 |
| supervised | zh | en | 30 | 49.713 | 64.980 | 70.107 | 65.850 | 78.295 | 82.265 |
| supervised | zh | en | 50 | 44.955 | 61.624 | 67.279 | 63.701 | 77.087 | 81.187 |
| supervised | zh | en | 70 | 38.687 | 57.405 | 63.815 | 61.068 | 75.854 | 80.121 |
| supervised | zh | en | 90 | 37.852 | 54.631 | 60.805 | 60.067 | 74.698 | 78.993 |

Table 13: Results on DBP15k dataset on experiments that uses me5 to generate the embeddings followed by MUSE alignment

| method | src_lang | tgt_lang | Hits@1 | Hits@10 | Hits@1 | Hits@10 |
|--------|----------|----------|--------|---------|--------|---------|
| unsupervised | fr | en | 95.183 | 98.941 | 97.243 | 99.609 |
| unsupervised | ja | en | 70.309 | 84.564 | 79.365 | 90.071 |
| unsupervised | zh | en | 51.129 | 72.020 | 64.999 | 82.456 |
| supervised | fr | en | 94.868 | 98.884 | 97.310 | 99.599 |
| supervised | ja | en | 68.803 | 83.193 | 79.672 | 90.061 |
| supervised | zh | en | 49.713 | 70.107 | 65.850 | 82.265 |

