# OpenReview forum: "Cross Lingual Long-tailed Entity Alignment in Knowledge Graphs"
_ICLR.cc/2026/Conference — Submitted to ICLR 2026_

### Official Review · Reviewer_a7nD · 2025-10-20

**Soundness:** 2
**Presentation:** 2
**Contribution:** 1
**Rating:** 2
**Confidence:** 5

**Summary:**

This paper addresses the long-tailed entity alignment (EA) problem. The authors propose ContrastEA, a novel approach that leverages pre-trained language models (e.g., mBERT) to generate entity representations, combined with a contrastive learning framework incorporating hard negative mining by selecting top-k negatives per entity and using the NT-Xent loss to better distinguish difficult entity pairs.

**Strengths:**

1. The paper introduces a novel dataset, which could be highly beneficial for the advancement of the field.

2. The writing is generally clear and coherent.

3. The paper addresses a novel problem and provides a preliminary solution.

**Weaknesses:**

1. The key techniques presented in the paper, like contrastive learning and hard sampling, have already been proposed in previous works like BootEA, Dual-AMN, and Lambda.

2. Aligning sparsely connected nodes is inherently challenging since there are no reference pre-aligned nodes to rely on; therefore, the alignment largely depends on leveraging the general knowledge encoded in large language models to enhance the initial entity representations. This approach is quite conventional and lacks novelty.

3. This paper can be regarded primarily as an engineering implementation, with its sole contribution being the introduction of a new dataset. Moreover, as the field evolves, knowledge graph technologies are increasingly being supplanted by large language models, which limits the practical applicability of the authors’ work.

4. The experimental results in the paper are disorganized and lack proper consolidation, indicating that the current version is far from being ready for publication.

**Questions:**

Refer to weaknesses. The authors are encouraged to provide effective responses.

---

> ### Author Response · Authors · 2025-12-03
> **Authors' Response (1/2)**
>
> We would like to thank the reviewer for his valuable feedback and hereby we address his concerns:
>
> W1 and W2: We would like to politely disagree with the reviewer's point of view here. We leveraged entity names and contrastive learning with hard sampling effectively as tools in this work for long-tailed entity alignment.
> Even though previous models have used these techniques, unlike ContrastEA,  these models are **entirely dependent on the triples and the relational information of the entities** in the KGs and exploit their **neighbourhood information** for the task of entity alignment.
> **BootEA** proposes a bootstrapping approach that iteratively labels entities to expand the training data to address the challenge presented by the lack of sufficient prior labelled training data for embedding-based EA. To sample hard negative examples, it introduces ϵ-truncated uniform negative sampling. It utilises Parameter Swapping to leverage prior alignment by generating supervised triples, calibrating embeddings in a unified space.
> **Dual-AMN** proposes a simplified KG encoder consisting of a Simplified Relational Attention Layer (capturing intra-graph information, i.e., triples) and a Proxy Matching Attention Layer (explicitly modelling cross-graph alignment via proxy vectors). It introduces the Normalized Hard Sample Mining Loss to generate hard negative samples.
>  **Lambda** uses a GNN-based encoder called KEESA (KG Entity Encoder with Selective Aggregation), which employs an adaptive dangling indicator applied globally for selective aggregation to prevent dangling entities from "polluting" matchable entity embeddings during neighborhood aggregation. It incorporates **an iterative method to detect the dangling entities as they are considered as noise**. Their **contrastive loss function disentangles matchable entities from dangling ones while learning a unified embedding space**.
>
> On the contrary, our **ContrastEA** model uses a tailored contrastive learning objective designed to **maximise separation in the embedding space**. The combination of top-k hard negatives with NT-Xent loss explicitly **increases the influence of negatives with large similarity, which helps separate challenging entity pairs**. KG structure-based models (like BootEA, Dual-AMN, and Lambda) rely on neighborhood information; since KGs are inherently sparse (often >50% of entities have ≤3 connections), these models struggle to capture true representations for sparsely connected entities. ContrastEA achieves robustness with entities with varying degrees, as shown in **Table 7** in the paper. Furthermore, we would like to highlight that ContrastEA overcomes the dependency on KG triples by relying solely on entity name representations generated by pre-trained Language Models (LMs), which are less sensitive to missing structural data. Furthermore, new results show that combining with entity descriptions improves the performance.  As a result, no training is required to generate the entity embeddings since they are derived from a pre-trained LM, making the model highly scalable to larger multilingual KGs. This contrasts with Dual-AMN and BootEA which require iterative graph operations, leading to be computationally expensive for larger KGs.
>
> Overall, our method relies on **entity names and type cues, leveraging them as robust features when relational triples are scarce**. This approach is purposefully designed for **aligning dangling and long-tailed entities**, which distinguishes it from prior works that assume richer relational structures. The model does not depend solely on LLMs but on a combination of name-based signals, which include type cues,  entity description (newly added feature) and contrastive learning, which together achieve strong alignment as shown in our experiments. We compared the results of BootEA for two benchmark datasets and Dual-AMN for one dataset, and the result shows that for Chinese dataset Dual-AMN works better, whereas for Japanese and French, ContrastEA outperforms them. There are no available results on Hits@k for Lambda, hence not included in the results.

---

> ### Author Response · Authors · 2025-12-03
> **Authors' Response (2/2)**
>
> W3: We respectfully disagree with the reviewer here and his comment on limited applicability. We would like to highlight the importance of our proposed framework and dataset.
>
> (1) Our work addresses a persistent and unsolved problem - long-tailed entity alignment. While LLMs have improved many NLP tasks, they do not replace the need for structured cross-lingual entity alignment, especially for long-tailed, low-frequency entities.  We conducted a few zero shot prompt-based experiments on Qwen-32B and DeepSeek-R1-70bn on randomly selected 1000 entity pairs from each of DE-EN, HI-EN, JA-EN, and AR-EN of our proposed LT-EA-25k dataset, and the results are given below for DeepSeek-R1-70bn, which is slightly better than Qwen-32b
> | Language Pair | Hits@1 |
> |---------------|--------|
> | DE-EN         | 19.2   |
> | HI-EN         | 12.1   |
> | JA-EN         | 10.4   |
> | AR-EN         | 11.8   |
>
> The limited performance infers that LLMs **cannot be used for cross-lingual entity alignment** with just the entity names in zero-shot settings. The prompts are provided in the updated manuscript.
>
> (2) Our methodology directly targets this under-served segment of the EA task by designing a framework that does not rely on relational triples and can align entities under minimal structural information, unlike the existing models.
>
> (3) Recent research (e.g., **retrieval-augmented LLMs, KG-RAG systems, hybrid neuro-symbolic architectures**) shows that structured KGs remain essential for factual accuracy, LLM outputs improve substantially when grounded in curated knowledge sources, and hallucination mitigation requires explicit entity-level grounding. KGs, being curated and structured, provide reliable facts that LLMs alone may not remember. Furthermore, unlike LLMs, KG can help in making reasoning transparent. Therefore, the importance of KGs is likely to go hand-in-hand as LLMs continue to advance.
>
>
> W4: We have organised the results in the manuscript for better understanding.

---

### Official Review · Reviewer_oU9k · 2025-10-27

**Soundness:** 2
**Presentation:** 2
**Contribution:** 3
**Rating:** 4
**Confidence:** 4

**Summary:**

This paper introduces ContrastEA, a novel contrastive learning-based model for cross-lingual entity alignment (EA) in knowledge graphs (KGs), with a particular focus on the long-tailed entities—those with sparse or no structural connections. The authors also construct a new multilingual benchmark dataset, LT-EA-25K, to better reflect real-world KG sparsity across seven languages. Experiments on several datasets demonstrate the effectiveness of the proposed method.

**Strengths:**

- The proposed LT-EA-25K dataset is a significant contribution, covering Arabic, German, Portuguese, Italian, Hindi, Russian, and Japanese aligned to English. It includes both long-tailed and dangling entities, which are often ignored in existing benchmarks.

**Weaknesses:**

- From my perspective, this paper does not present significant technical innovation. The use of negative sampling and contrastive learning frameworks has already been explored in prior work.

- Regarding the motivation, the paper focuses on long-tailed entities, yet the proposed method does not seem to directly address the core challenge, i.e., the lack of relational triples for long-tailed entities. Although the method incorporates name-based features, in my experience, name alone can resolve most entity alignment cases. In fact, the experimental results show strong performance on long-tailed datasets, but it remains unclear how much of that success is attributable to the name feature. Overall, there is a disconnect between the method and its stated motivation, and the key contributing module behind the experimental gains is not clearly identified.

**Questions:**

- What about the performance of the method that only uses name embeddings for entity alignment on your dataset?

---

> ### Author Response · Authors · 2025-12-03
> **Authors' Response**
>
> We would like to thank the reviewer for their feedback on the paper and hereby address their concerns
>
> W1: While negative sampling and contrastive learning have been explored in prior work, in our paper, we use them as tools to enable effective long-tailed entity alignment. In this paper, our key contribution is to leverage these techniques and apply them to the entity alignment task, specifically on datasets which contain long-tailed entities across low resource languages. Negative sampling plays an important role here because it allows the model to focus on hard negatives, i.e., entities that are similar in name or type cues in the names, which is crucial for disambiguating ambiguous entities. Contrastive learning provides a flexible framework to align entities across languages by pulling together positive pairs and pushing apart negatives, which is highly effective for datasets with sparse training signals for long-tailed entities.  Moreover, the results on the existing benchmark datasets SRPRS, DBP15k, and DBP5L given in Tables 3,4, and 5 in the manuscript show that our proposed entity alignment model outperforms the existing baseline models across all datasets and achieves comparable results with ZH-EN language pair. However, it is to be noted that all these existing models, except DAT use triples and graph structural information for EA task, unlike ContrastEA.
>
> W2. Due to the lack of relational triples for long-tailed entities, we rely primarily on entity names for alignment. Since this work do not focus on knowledge graph completion, we do not deal with fixing the missing relational information. By sampling hard negatives that are structurally and semantically similar to the anchor entity, the model learns to disambiguate entities even when relational triples are scarce. This is especially important for long-tailed entities where neighbors provide limited information. The results shown in Table 6 (in black) in the paper are the results of the entity alignment when only names are used. With further experiments (in the table above), we can conclude that entity names together with the type cues are more beneficial compared to entity descriptions as they help with disambiguation.
>
> Q1. We are unclear about this question from the reviewer.  If he refers to the results of our ContrastEA method with names, then they were already provided in the paper in Table 6. If he is referring to any other model which uses name embeddings e.g., DAT model and the performance of the DAT model on our LT-EA-25k dataset, then we again would like to refer to Table 6, where we showed the results from the DAT model on our dataset, which uses the name embeddings.

---

### Official Review · Reviewer_qm2m · 2025-10-30

**Soundness:** 2
**Presentation:** 2
**Contribution:** 3
**Rating:** 4
**Confidence:** 4

**Summary:**

This paper introduces ContrastEA, a contrastive-learning-based approach for cross-lingual long-tailed entity alignment that relies purely on entity names encoded by a multilingual sentence encoder. The method uses top-k hard negative mining and the NT-Xent loss to enhance discrimination, aiming to tackle alignment under long-tailed and dangling-entity conditions where structural or relational signals are sparse.
In addition, the authors construct LT-EA-25K, a new benchmark covering 7 language pairs with realistic long-tailed degree distributions and dangling entities. Experiments show that ContrastEA achieves strong or state-of-the-art results over a structure-based baseline.

**Strengths:**

1. The experiments and analysis demonstrate that relying solely on entity names circumvents structural sparsity in long-tailed KGs while still delivering competitive alignment accuracy.

2. The top-k strategy under NT-Xent is intuitive and empirically useful for emphasizing informative negatives.

3. LT-EA-25K provides a much-needed long-tailed, multilingual dataset including dangling entities, making it valuable to the community.

**Weaknesses:**

1. As ContrastEA just utilizes the entity name for EA, the paper needs to deeply analyze failure modes such as homonyms, transliteration mismatches, or domain-specific ambiguities like same names referring to different entities. It would be better if you could quantify the limitations and provide results for same-name collisions and one-sentence description augmentation.

2. Lack of sensitivity to k, τ, and batch size is not explored. All negatives are selected within the mini-batch. This can lead to false negatives and heavy dependence on batch size.

3. Reported numbers lack standard deviations or significance testing. Without error bars or paired t-tests, it is unclear whether improvements are statistically meaningful.

4. It seems this paper misses preprocessing details. Name normalization strongly affects name-only alignment but is not described. It would be better if you could provide them.

5. Although the paper has claimed efficiency, the paper reports no throughput or GPU memory statistics. Top-k hard-negative mining scales as O(batch²). I suggest adding runtime and memory benchmarks.

**Questions:**

Please see the weaknesses.

---

> ### Author Response · Authors · 2025-12-03
> **Authors' Response**
>
> We thank the reviewer for their insightful comments and hereby we address their concerns.:
>
> W1: We agree with the reviewer that entity descriptions might be helpful for long-tailed entities and for different languages. We have conducted new experiments, and the results are updated in the manuscript in Table 6 and also provided in the response for the reviewer oYhE
>
> W2: We did an analysis on the sensitivity of k and batch size, and the results are provided in the updated manuscript. The results indicate that the model is quite robust to batch size changes—performance does not fluctuate drastically. The model is also robust to a range of top_k values. Using top_k = 1–3 is sufficient and slightly better for maximising Hits@1, while larger top_k values offer diminishing returns. Therefore, hard negatives help in better performances.
>
> W3: We have updated each results table (Table 3, 4 and 5) with the improvement of ContrastEA w.r.t each baseline model and is color coded.
>
> W4. The only preprocessing step applied to entity names was URL cleaning. For e.g., https://dbpedia.org/page/Paris_(mythology) was converted to Paris (mythology) to obtain the clean entity name. No additional normalisation (e.g., lowercasing, accent removal, stemming) was performed. We provided this information in the updated manuscript.
>
> W5. We have provided a detailed analysis of this in the updated manuscript.

---

### Official Review · Reviewer_oYhE · 2025-10-31

**Soundness:** 3
**Presentation:** 2
**Contribution:** 3
**Rating:** 4
**Confidence:** 3

**Summary:**

This paper addresses the challenge of aligning sparsely connected long-tailed entities in cross-lingual knowledge graphs (KGs), where traditional entity alignment (EA) models underperform due to insufficient structural and neighbourhood information. The authors propose ContrastEA, a novel model that leverages pre-trained multilingual language models (e.g., mE5) to generate entity representations and incorporates a contrastive learning framework with hard-negative mining (top-k negatives per entity) and NT-Xent loss to distinguish similar entities. Additionally, to address the under-representation of long-tailed entities in existing benchmarks, the authors curate a new dataset LT-EA-25K from DBpedia, covering seven languages (Arabic, German, Portuguese, Italian, Hindi, Russian, Japanese) aligned to English, including 154,296 cross-lingual entity pairs and dangling entities (zero connections). Experimental results show that ContrastEA outperforms classic and long-tailed EA models on three benchmark datasets (DBP15K, SRPRS, DBP5L) by improving Hits@1 by 6–20 percentage points and achieves state-of-the-art (SOTA) performance on LT-EA-25K. The paper’s key contributions include the ContrastEA model, the LT-EA-25K dataset, and empirical validation of robustness across entities with varying connectivity and diverse language pairs.

**Strengths:**

1. The work identifies a critical gap in existing EA research—neglect of long-tailed and dangling entities in real-world KGs—and proposes a targeted solution that abandons reliance on structural/relational information, a bold and effective design choice given the sparsity of such entities.
2. ContrastEA demonstrates strong originality by combining pre-trained multilingual LMs with hard-negative mining and NT-Xent loss; the top-k hard negative strategy addresses the limitation of uninformative negative samples in prior contrastive learning for EA, enhancing the model’s ability to distinguish deceptive similar entities.
3. The curation of LT-EA-25K fills a crucial gap in benchmark datasets, as it reflects real-world KG sparsity (including entities with degree ≤3 and dangling entities) and covers low-resource languages (e.g., Hindi, Arabic), making it a valuable resource for future long-tailed EA research.

**Weaknesses:**

1. The model relies solely on entity names for representation, ignoring other potentially useful textual features (e.g., short entity descriptions or attributes) that may exist for some long-tailed entities. For entities with ambiguous names (e.g., homonyms across languages), integrating such supplementary information could further improve alignment accuracy, especially for low-resource languages with sparse name data.
2. The paper does not explore the impact of different pre-trained LMs on model performance.
3. The LT-EA-25K dataset’s construction process, while detailed, does not include a quantitative analysis of label quality. For example, how were the owl:sameAs links verified for low-resource languages with limited annotation resources? Additionally, the dataset’s generalizability to non-DBpedia KGs (e.g., Wikidata, YAGO) is not evaluated, as experiments are limited to DBpedia-derived benchmarks.

**Questions:**

1. How does the model handle cases where the top-k hard negatives include entities from the same semantic category as the anchor (e.g., two different "Paris" entities in different languages)? Does the current framework effectively distinguish such fine-grained semantic differences, or could a category-aware negative sampling approach further improve performance?
2. What proportion of entities in each language pair are true long-tailed (degree ≤3) versus dangling (degree=0)? How does ContrastEA’s performance vary specifically on dangling entities compared to entities with 1–3 connections? Providing disaggregated results would clarify the model’s ability to handle the most extreme sparsity.

---

> ### Author Response · Authors · 2025-12-03
> **Authors' Response (1/2)**
>
> We thank the reviewer for his insightful feedback and comments. We address the concerns below:
>
> W1: We agree with the reviewer that entity descriptions might be beneficial and hence conducted additional experiments on our LT-EA-25k dataset using 1-sentence entity descriptions wherever available to analyse if they are useful for long-tailed entities in low-resource languages. The results are presented in the table below (new results are in bold and italics), revealing several important trends.
> ### Results for the LT-EA-25k Dataset
> | **Language Pair** | **Hits@1** | **Hits@3** | **Hits@10** | **DAT Hits@1** | **DAT Hits@10** |
> |-------------------|-----------|-----------|-------------|----------------|-----------------|
> | **AR-EN (Name)** | 81.42 | 88.39 | 92.67 | 1.44 | 4.78 |
> | ***AR-EN (Desc)*** | ***58.17*** | ***63.19*** | ***66.81*** |  |  |
> | **HI-EN (Name)** | 86.17 | 92.82 | 95.55 | 1.36 | 6.56 |
> | ***HI-EN (Desc)*** | ***87.23*** | ***94.59*** | ***96.55*** |  |  |
> | **IT-EN (Name)** | 95.72 | 96.91 | 97.67 | 0.75 | 3.93 |
> | ***IT-EN (Desc)*** | ***86.14*** | ***89.35*** | ***92.53*** |  |  |
> | **RU-EN (Name)** | 90.12 | 94.10 | 96.09 | 0.56 | 2.68 |
> | ***RU-EN (Desc)*** | ***87.54*** | ***92.57*** | ***95.83*** |  |  |
> | **JA-EN (Name)** | 53.33 | 59.73 | 65.95 | 0.97 | 3.40 |
> | ***JA-EN (Desc)*** | ***83.60*** | ***89.22*** | ***92.79*** |  |  |
> | **PT-EN (Name)** | 96.11 | 97.84 | 98.38 | 1.05 | 4.60 |
> | ***PT-EN (Desc)*** | ***94.74*** | ***97.06*** | ***98.15*** |  |  |
> | **DE-EN (Name)** | 97.31 | 98.49 | 99.05 | 0.59 | 3.39 |
> | ***DE-EN (Desc)*** | ***93.74*** | ***96.10*** | ***97.46*** |  |  |
>
> Our findings from these new results are as follows:
>
> 1. We observe the most notable gain for ***Japanese***, where ***Hits@1 jumps from 53.33 → 83.60***. This supports the hypothesis that Japanese entities often have short or transliterated surface forms that provide very weak semantic signals, whereas even brief descriptions supply rich contextual cues
> For e.g. the entity 白い暴動 (アルバム) translates to White Riot (album) and our prediction model with entity names mapped it to the White Riot entity from English DBpedia, but the correct corresponding English entity is The Clash (album).
> But, when we use sentence description instead of the entity name, it could correctly map it to the English entity is The Clash (album). Therefore, descriptions substantially improve alignment for low-resource languages or script-divergent languages. It also shows mild improvements in Hindi as well.
>
> 2. The entity names in DBpedia often contain the relevant disambiguation information, for instance, ***The Clash (album)***. This entity type information embedded in the entity names proves to be beneficial for entity alignment. This can be concluded from the table above, for e.g., Arabic shows a significant degradation (81.42 → 58.17 in Hits@1). We inspected this phenomenon closely and found that for entities like “حرب النجوم (فيلم)” -> ‘Star Wars (film)’ got correctly aligned when only names are used. When switching to descriptions alone, this strong built-in disambiguation signal is lost, leading to poorer performance. In such cases, the name itself is more informative than the short textual description. Similar observations are also made for Italian, Russian and German entities.
>
> 3. Motivated by these new findings, we further ran experiments combining both names and descriptions and the results are provided in the table below:
> | Language Pair          | Hits@1 | Hits@3 | Hits@10 |
> |------------------------|--------|--------|---------|
> | AR–EN (Name+Desc)      | 83.33  | 89.10  | 93.14   |
> | HI–EN (Name+Desc)      | 88.20  | 95.20  | 96.80   |
> | IT–EN (Name+Desc)      | 96.14  | 97.05  | 97.91   |
> | RU–EN (Name+Desc)      | 90.74  | 94.87  | 96.88   |
> | JA–EN (Name+Desc)      | 85.23  | 92.56  | 93.87   |
> | PT–EN (Name+Desc)      | 96.77  | 97.90  | 98.72   |
> | DE–EN (Name+Desc)      | 97.90  | 98.72  | 99.23   |
>
> The results show that the combination of entity names with embedded entity type information, together with short entity descriptions, improves the proposed model ContrastEA. The new results are provided in Table 6 in the paper.
>
> W2: We appreciate the reviewer’s suggestion to explore the impact of different pre-trained language models. While we agree that this is an important dimension, we emphasised designing an architecture which is model-agnostic and allows different pre-trained language models to be plugged in with minimal to no adjustments, because to keep up with the rapidly growing space of the pretrained LMs, we need a framework to accommodate all. This extensibility allows for the seamless integration of alternative encoders by future researchers or practitioners with domain-specific needs, facilitating the examination of their effects on performance. We clarified this motivation in the revised version of the paper.

---

> ### Author Response · Authors · 2025-12-03
> **Authors' Response (2/2)**
>
> We continue with our answers on the weaknesses and the questions here:
>
> W3: (i) Quality of owl:sameAs links: The owl:sameAs links in LT-EA-25K were directly extracted from DBpedia. These links are curated and maintained by the DBpedia Association, and therefore, we did not perform additional manual validation.
> (ii) generalizability beyond DBpedia (e.g., Wikidata, YAGO): Our dataset specifically targets cross-lingual entity alignment across different DBpedia language editions. We did not include Wikidata because Wikidata is a single, language-independent KG where entities are uniquely identified (e.g., Q937 for Albert Einstein), and multilinguality appears only at the level of labels/surface forms. Consequently, standard cross-lingual entity alignment within the entities in Wikidata is not applicable in this context, since all language versions map to the same entity ID by design. However, the dataset construction methodology is not limited to DBpedia. It can be extended to YAGO by following the same pipeline, and we will highlight this in the revised manuscript.
> (iii) Experiments with other  KGs: We did not evaluate ContrastEA on cross-KG benchmarks such as DBP-YAGO or DBP-Wikidata because existing benchmarks (e.g., SRPRS) use English-only surface forms, and therefore do not align with the multilingual focus of our work.
>
> Q1: In DBpedia, entity types are often implicitly encoded in the surface forms themselves (e.g., “Paris (city)”, “Paris, Texas”, “Paris (mythology)”), and this linguistic signal is captured in the entity embeddings generated using me5. When generating top-k hard negatives, our intuition is that these entity type cues in the embedding space influence which candidates are selected as negatives. We also observe this indirectly in our experiments: for example, the performance with entity descriptions drops sharply for Arabic compared to using entity names with the type cues. This supports the claim of the paper that the disambiguation power comes largely from surface-form signals. For long-tailed entities, however, the DBpedia ontology assigns the majority to the very general class owl: Thing, which does not meaningfully aid disambiguation. Because type information is sparse or uninformative for these entities, we did not incorporate explicit category-aware or type-constrained negative sampling in ContrastEA. However, for the popular DBpedia entities, such a category-aware strategy could indeed provide additional benefits. We updated the manuscript with this information accordingly.
>
> Q2: The degree distribution is already provided in the Figure 1 and Table 7 (in appendix)  and Hits@k w.r.t degrees are provided in Figure 2. As mentioned in the paper, there are 1400 dangling entities in the ar-en language pair. Overall, aggregated result of degree = 0, degree≤3 and rest with entity names are given below:
>   | Language | Group | Hits@1 | Hits@3 | Hits@10 |
> |------|--------|---------|---------|-----------|
> | AR-EN | Deg 0 | 0.6182 | 0.6435 | 0.687 |
> | AR-EN | Deg 1–3 | 0.8694 | 0.9305 | 0.9613 |
> | AR-EN | Deg 4–>5 | 0.8556 | 0.9099 | 0.9454 |
> | HI-EN | Deg 1–3 | 0.9062 | 0.9548 | 0.9777 |
> | HI-EN | Deg 4–>5 | 0.8876 | 0.9402 | 0.9737 |
> | JA-EN | Deg 1–3 | 0.6489 | 0.7219 | 0.7902 |
> | JA-EN | Deg 4–>5 | 0.5769 | 0.6381 | 0.7109 |
> | PT-EN | Deg 1–3 | 0.9857 | 0.9932 | 0.9957 |
> | PT-EN | Deg 4–>5 | 0.9719 | 0.9775 | 0.9797 |
> | RU-EN | Deg 1–3 | 0.9471 | 0.9732 | 0.9850 |
> | RU-EN | Deg 4–>5 | 0.9243 | 0.9535 | 0.9669 |
>
>
> For all languages, the Hits@k metrics and Accuracy are higher for entities with degrees 1–3 compared to entities with degrees 4, 5, and >5. This indicates that entities with lower connectivity in the KG (degree 1–3) are easier to align, likely because they are less entangled with other entities and have fewer potential hard negatives. However, popular entities containing more information tend to be comparatively ambiguous. Also, Japanese suffers the most with high-degree entities due to short/transliterated surface forms and low-resource characteristics.
>
> Other languages  (PT, RU, HI) maintain high performance, showing the robustness of the model when enough disambiguating information exists. This result has also been provided in the updated manuscript.

---

### Meta-Review · Area_Chair_P7U1 · 2026-01-07

**Summary:**

This paper studies entity alignment for long-tailed and dangling entities and proposes ContrastEA, a contrastive learning framework based on textual entity representations with top-k hard-negative mining, together with a new multilingual dataset, LT-EA-25K.

Reviewers consistently agree that LT-EA-25K is a strong and valuable contribution, filling an important gap in existing EA benchmarks by explicitly modeling long-tailed and dangling entities. The modeling choice to abandon structural signals is well motivated for this setting, and the top-k hard-negative strategy is intuitive and empirically effective.

However, several reviewers raise concerns about limited technical novelty, noting that contrastive learning and hard-negative sampling have been explored in prior EA work. There is also a perceived disconnect between the stated motivation (long-tailed sparsity) and the method, as strong performance may largely stem from entity names and pretrained LMs. The paper further lacks key analyses, including failure modes of alignment, statistical significance, and disaggregated results for dangling versus low-degree entities.

Overall, while the dataset contribution is compelling and the empirical results are promising, the current submission would benefit from stronger analysis and clearer positioning of its methodological contribution.

**Reviewer Concerns:**

see metareview

**Reviewer Scores:**

see metareview

---

### Decision · Program_Chairs · 2026-01-26

Reject